# Research on Design and Analysis Method of the Double Planetary HMCVT Based on High Efficiency Transmission

**Jiang Li [1,2], Yirong Zhao [3], Zhiqiang Zhai [1,2], Bing Han [1,2], Yuefeng Du [1,2], Lin Wang [3] and Zhongxiang Zhu [1,2,*]**

1   College of Engineering, China Agricultural University, Beijing 100091, China
2   Beijing Key Laboratory of Optimized Design for Modern Agricultural Equipment, Beijing 100091, China
3   State Key Laboratory of Power System of Tractor, Luoyang 471039, China
*   Correspondence: zhuzhonxiang@cau.edu.cn

**Abstract:** The hydro-mechanical continuously variable transmission (HMCVT) is the critical component in the transmission system of the high-horsepower tractor. However, different structural layouts have a significant influence on transmission efficiency, especially for the HMCVT with multiple planetary rows. Therefore, the double planetary HMCVT, which, based on efficiency characteristics, was designed in this study, as well as the efficiency distribution area under different loads, were analysed. First, the method of the single planetary row in structure layout is constructed and redefined, revealing the transformation law of transmission in different layouts. Moreover, the different layout efficiencies and output transmission ratios were derived as the theoretical basis for selection. Then, the obtained transmission and efficiency characteristics were selected as the best combination to design the double planetary HMCVT. The theoretical efficiency and hydraulic power shunt characteristics were analysed with the circulating power, and the influence of circulating power on operational efficiency was determined. Finally, the hardware-in-the-loop system of HMCVT was designed. Herein, a new type of variable load efficiency characteristic analysis method is proposed, which treats the engine and transmission as an efficient whole. The variational performance of the efficiency field in hydro-mechanical stages are discussed under full load and variable load. This research provides theoretical support for efficiency improvements in design and analysis of the multi-planetary HMCVT tractor.

**Keywords:** hydro-mechanical continuously variable transmission; transmission efficiency; variable load efficiency field; power split; hardware-in-the-loop system



## 1. Introduction

With the rapid development of smart agriculture [1], tractors play an important role in agricultural development as the power machinery in agriculture. The hydro-mechanical continuously variable transmissions (HMCVT) are widely used in heavy agricultural machinery, because the hydro-mechanical transmission has better driving comfort compared with traditional transmissions [2]. Moreover, the HMCVT can match the engine with the best transmission efficiency and fuel economy [3]. Depending on the composition of structure, the HMCVT can be divided into input and output coupling [4]. The connection between planetary gears and drive shafts generates six different layouts, each of which has different performance and efficiency. For high-horsepower tractors, the double planetary row structure of the HMCVT is generally a combination of two layouts, so the structure of the double planetary row has 36 combinations, and different combinations have different efficiency and transmission characteristics [5].

Some scholars have researched the layout of hydraulic–mechanical transmission and efficiency characteristics. The steady-state numerical multi-range hydro-mechanical transmission (HMT), which is based on a dual-stage input coupled layout was established, and the speed, power flow and efficiency of the split–continuous variable transmissions were

analysed by a modular design method [6]. Antonio and Alarico [7] studied the efficiency issues of engines equipped with HMCVT and proposed an engine–HMCVT joint control strategy, which improved the fuel economy and transmission efficiency of the entire system. Then, Antonio and Alarico [8] proposed the scheme of three-axis hydro-mechanical transmission in HMT, and took it as an efficiency matching and transmission ratio optimization problem, which was verified by a 75 kW forklift. The fast kinematics method was proposed to be suitable for the identification of the functional parameters in the compound power split continuously variable transmission, and the method showed that the actual gear set layout had efficiency as a priority in comparison with the transmission ratio [9]. The static characteristics of different forms of hydro-mechanical power split gearboxes were analysed, and two methods were proposed to improve the fuel efficiency of vehicles: one method was to improve the efficiency of the engine, and the other was to change the transmission layout of the gearbox for the purpose of improving the matching degree of the system [10]. Considering the optimisation of transmission efficiency, Zhun Cheng [11] proposed a new HMCVT transmission parameters optimisation method that improved the transmission parameters by the genetic algorithm and determined the most appropriate point of the intersection in each gear stage. The results indicated that the optimized HMCVT had better transmission efficiency and enhanced practicality. Yu Xia et al. [12] studied the transmission characteristics of single-row power split hydro-mechanical continuously variable transmission (PSHMCVT), and optimized the parameters of the structure layout. The results showed that the optimized speed ratio and torque ratio increased by 32.27% and 70.22%, respectively. However, the average efficiency declined.

The above research mainly focused on analyses of existing gear structures and operating conditions under standard load. There is no specific analysis on the reasons for the layout design of HMCVT, and there is less research on multi-planetary row and variable load. This article analysed the layout of the single planetary row of HMCVT and showed the potential transmission characteristics between the different layouts. The layouts were mapped to the mathematical formulas with speed and torque as the standard. Then, according to the efficiency characteristics of different layouts and the transmission ratio conditions, the appropriate layouts were selected and combined to design the dual planetary HMCVT. Finally, the hardware-in-the-loop system of the double planetary HMCVT was designed, and the HMCVT, the engine and the load were used as a system to analyse the efficiency characteristics of the tractor in full load with the power split characteristics. Moreover, the efficiency field of the variable load is discussed in HM1 and HM2.

## 2. Materials and Methods

### 2.1. Design of Double Planetary HMCVT

2.1.1. Functional Design of Different Layouts

The speed equation of a simple planetary gear is expressed as a function, which is shown as the characteristic parameter $k$ of the planetary gear [13].

$$n_s + kn_r - (1+k)n_c = 0 \tag{1}$$

Similarly, the torque equation of the planetary gear is expressed as a function of the characteristic parameter $k$ in the planetary gear in ideal conditions [14].

$$T_s : T_r : T_c = 1 : k : -(1+k) \tag{2}$$

where $n$, $T$ and $k$ represent the rotational speed, torque and planetary row structure parameters, respectively; the subscripts $s$, $r$ and $c$ represent the sun gear, ring gear and planetary carrier, respectively.

Equations (1) and (2) define a plane in a three-dimensional space. The three points belonging to the plane can be used to express the relationship of rotation speed and torque, respectively. In this study, these points are defined by selecting a reference axis. The constant speed of the reference axis is assumed to be 1, and the speed of the remaining axis

is calculated when the other axis is stationary. Assuming that the planet carrier is used as a reference, Equation (1) can be represented by the three points in Table 1.

**Table 1.** Definition points of *crs* layout in $(n_s, n_c, n_r) \in \mathbb{R}^3$.

| $Q_{crs}$ | $n_c$ | $n_r$ | $n_s$ |
|---|---|---|---|
| $q_0$ | 0 | 0 | 0 |
| $q_1$ | 1 | 0 | $1+k$ |
| $q_2$ | 1 | $(1+k)/k$ | 0 |

All the null values and relative unit values are ignored in Table 1, and the three points $(q_0, q_1, q_2)$ can be reduced to only one point in the two-dimensional space, tracking the order used for notation:

$$Q_{crs} = [(1+k)/k; 1+k] = \left[ \frac{n_r}{n_c} \Big|_{n_s=0}; \frac{n_s}{n_c} \Big|_{n_r=0} \right] \tag{3}$$

where the first letter of the subscript of $Q$ is the speed of the reference shaft (planet carrier), and the second and third letters represent the speed of the corresponding ring gear and sun gear. $n_r$, $n_s$ and $n_c$ represent the rotating speed of the gear ring, sun gear and planet carrier, respectively.

The same form of expression is where $a$, $b$ and $c$ represent the three components of the planetary gear mechanism. The general representation of $Q$ is:

$$Q_{abc} = [x_1; y_1] = \left[ \frac{\omega_b}{\omega_a} \Big|_{\omega_c=0}; \frac{\omega_c}{\omega_a} \Big|_{\omega_b=0} \right] \tag{4}$$

where $x_1$ represents the speed ratio of the *b*-axis and *a*-axis when $\omega_c = 0$; $y_1$ represents the speed ratio of the *c*-axis and *a*-axis when $\omega_b = 0$; and $\omega_a$, $\omega_b$ and $\omega_c$ represent the angular velocity of shaft the connecting the component (*a*), the angular velocity of the shaft connecting the component (*b*) and the angular velocity of the shaft connecting the component (*c*).

The different layouts are converted, and the form of planetary gears (subscript *a b c*) can be shown in six forms: *crs*, *csr*, *scr*, *src*, *rcs* and *rsc* [4]. The speed and torque distributions of the six layouts is shown in Table 2.

**Table 2.** Different layout of corresponding values of speed and torque when $(x; y) \in \mathbb{R}^2$.

| Layout (Speed) | $x_1$ | $y_1$ | Layout (Torque) | $x_2$ | $y_2$ |
|---|---|---|---|---|---|
| $Q_{scr}$ | $1/(1+k)$ | $-1/k$ | $T_{scr}$ | $-(1+k)$ | $k$ |
| $Q_{rcs}$ | $k/(1+k)$ | $-k$ | $T_{rcs}$ | $-(1+k)/k$ | $1/k$ |
| $Q_{crs}$ | $(1+k)/k$ | $1+k$ | $T_{crs}$ | $-k/(1+k)$ | $-1/(1+k)$ |
| $Q_{src}$ | $-1/k$ | $1/(1+k)$ | $T_{src}$ | $k$ | $-(1+k)$ |
| $Q_{rsc}$ | $-k$ | $k/(1+k)$ | $T_{rsc}$ | $1/k$ | $-(1+k)/k$ |
| $Q_{csr}$ | $1+k$ | $(1+k)/k$ | $T_{csr}$ | $-1/(1+k)$ | $-k/(1+k)$ |

The twelve two-dimensional points are all expressions of different layouts, according to Equations (1) and (2). It can be seen from Table 2 that the connection between the planetary gear shaft and the drive shaft will affect the overall transmission performance. Therefore, it is necessary to analyse the shaft connecting the different parts of the planetary gear. The three main components of the planetary gear are the sun gear, the ring gear and the planet carrier. On the system side, the functions of the shafts or the main components which they drive can be used to identify the shafts. According to the transmission characteristics of the HMCVT, the three shafts are, respectively, the shaft connected to the engine (ICE), the shaft connected to the pump-motor hydraulic system (PM) of the HMCVT and the shaft connected to the output shaft (OUT) of HMCVT.

The layout of the planetary gear train and the characteristic parameters of the planetary gear in the HMCVT have a great influence on the transmission characteristics; thus, it is of great significance to the design and selection of the planetary gear layout [15]. According to Table 2, the six different layouts are the functions of the planetary gear characteristic parameter $k$. Based on experience, the value of the planetary gear characteristic parameter $k$ is between 1 and 10. Therefore, the layout characteristics are interpreted as mathematical formulas, as shown in Equations (5) and (6).

$$C \in \mathbb{R}^2, C = f(\text{layout}, k), \text{layout} \in (crs, csr, scr, src, rcs, rsc), k \in (1, 10)$$

$$Q \in C, Q(x_1, y_1) = f(layout, k) = \begin{cases} [(1+k)/k, 1+k], \text{layout} = crs \\ [1+k, k/(1+k)], \text{layout} = csr \\ [1/(1+k), -1/k], \text{layout} = scr \\ [-1/k, 1/(1+k)], \text{layout} = src \\ [k/(1+k), -k], \text{layout} = rcs \\ [-k, k/(1+k)], \text{layout} = rsc \end{cases} \quad (5)$$

$$T \in C, T(x_2, y_2) = g(layout, k) = \begin{cases} [-k/(1+k), -1/(1+k)], \text{layout} = crs \\ [-1/(1+k), -(1+k)/k], \text{layout} = csr \\ [-(1+k), k], \text{layout} = scr \\ [k, -(1+k)], \text{layout} = src \\ [-(1+k)/k, 1/k], \text{layout} = rcs \\ [1/k, -(1+k)/k], \text{layout} = rsc \end{cases} \quad (6)$$

where $C$ represents a set of two-dimensional numbers and $\mathbb{R}^2$ represents the two-dimensional space.

### 2.1.2. The Representation of the Different Layouts

In order to express the transmission characteristics of six different layouts and to provide design solutions for HMCVT, this article defines the transmission shaft which connects to the planetary gear mechanism: the input shaft of the engine is used as the reference shaft, and its speed is kept at a fixed value ($a$). The shaft connected to the pump-motor hydraulic system of the HMCVT and the shaft connected to the output rear axle of the vehicle are referred to as $b$ and $c$, respectively. Equation (4) can be expressed as:

$$Q_{abc}(x_1, y_1) = [\frac{b}{a}|_{c=0}, \frac{c}{a}|_{b=0}] = [\frac{\omega_{PM}}{\omega_{DE}}|_{\omega_{OUT}=0}, \frac{\omega_{OUT}}{\omega_{DE}}|_{\omega_{PM}=0}] \quad (7)$$

where $x_1$ represents the zero-speed point, and $y_1$ represents the full mechanical point; $\omega$ represents the angular velocity; the subscripts *PM*, *DE* and *OUT* represent the pump-motor hydraulic system of the HMCVT, the diesel engine and the output shaft of the HMCVT, respectively.

According to the above connection rules, there are six different layout forms which are shown in Table 3. The six different layouts *xyz* (*scr*, *src*, *rsc*, *rcs*, *crs*, *csr*) can be described as the following: the output shaft of the engine is connected to *x* component; the pump-motor hydraulic mechanism of the HMCVT is connected to *y* component, and the output shaft of HMCVT (the input shaft of the rear axle) is connected to *z* component, respectively. For example, the layout *scr* can be described as follows: the output shaft of the engine is connected to the sun gear; the pump-motor hydraulic mechanism of the HMCVT is connected to the planet carrier and the output shaft of HMCVT (the input shaft of the rear axle) is connected to the ring gear, respectively.

**Table 3.** Transmission schemes with different layouts.

| Layout | Transmission Scheme | Layout | Transmission Scheme |
|---|---|---|---|
| *scr* | | *src* | |
| *rsc* | | *rcs* | |
| *crs* | | *csr* | |

### 2.1.3. Analysis of Transmission Ratio and Efficiency Characteristics in Different Layouts

$v_r$ is defined as the ratio of the output speed of the vehicle to the maximum vehicle speed.

$$v_r = \frac{V_{\text{ove}}}{V_{\text{max}}} \tag{8}$$

According to Equations (7) and (8), the ratio between output speed of the vehicle and output speed of the HMCVT is shown as follows:

$$i_{\text{out}} = \frac{\omega_{RA}}{\omega_{OUT}} = \frac{V_{ove}}{2\pi r_w \omega_{DE}} \frac{\omega_{DE}}{\omega_{OUT}} = \frac{v_r V_{\text{max}}}{2\pi r_w \omega_{DE}} \frac{1}{y_1} \tag{9}$$

where $i_{\text{out}}$ represents transmission ratio of vehicle output shaft; $\omega_{RA}$ represents the angular velocity of vehicle wheels; $\omega_{OUT}$ represents output the angular velocity of the HMCVT and $r_w$ represents the radius of the vehicle wheel.

It can be intuitively seen that $i_{\text{out}}$ depends on $v_r$ and $y_1$. The $v_r$ is the ratio of the output speed to the maximum vehicle speed, which ranges from 0.1 to 1. It can be seen from Figure 1 that $i_{\text{out}}$ has different value ranges for the six structural layout types of *csr*, *scr*, *scr*, *src*, *rcs* and *rsc*, which change continuously with the value of *k* and $v_r$. However, the value of $i_{\text{out}}$ is less than 1, because the function of the gearbox reduces the speed and increases the torque to configure the requirements of low-speed and high-horsepower for tractors. It can be seen from Figure 1 that the transmission ratio $i_{\text{out}}$ under the *scr* and *src* layouts are larger than the other four layouts, and they are greater than 1, so the performance when increasing the transmission torque is the worst under these two layouts. The other four layouts (*csr*, *crs*, *rcs* and *rsc*) have low transmission ratios, and their influence on the reduction of rotational speed and increasing torque are relatively noticeable. Therefore, *csr*, *crs*, *rcs* and *rsc* can each be used as one of the indicators for selecting the structure of HMCVT.

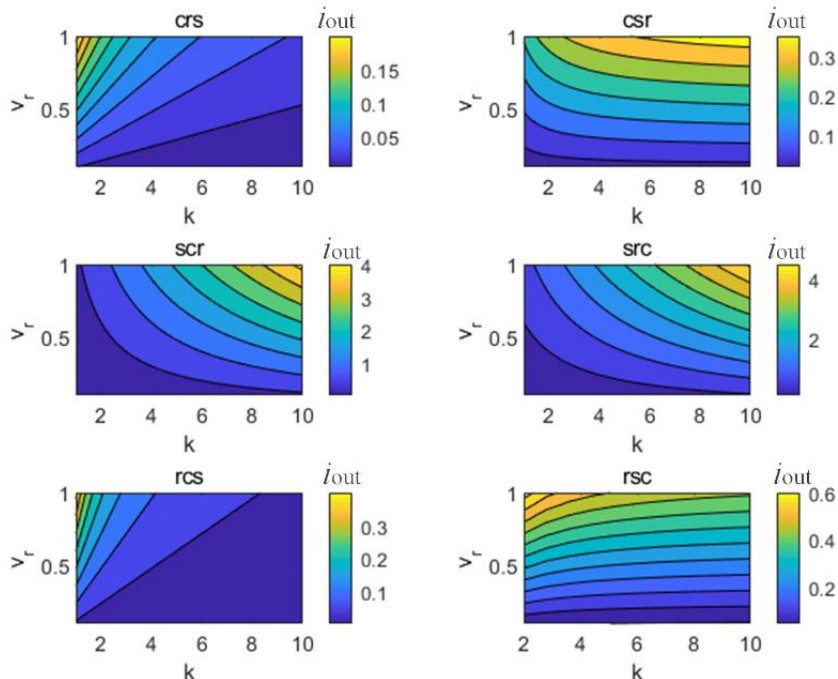

**Figure 1.** Result distribution graph of $i_{out}$.

In order to select the appropriate combination mode, the influence of the structure layout on the transmission efficiency is explored. A 200-horsepower tractor was selected as the design vehicle. The parameters of the tractor are reported in Table 4. The efficiency of the pump motor hydraulic system is calculated as follows [16]:

$$\eta_{pm} = \eta_p \eta_m = \frac{1 - C_s \dfrac{\Delta P_p}{|e|\mu n_p}}{1 + C_V \dfrac{\mu n_P}{10\Delta P_p |e|} + \dfrac{C_f}{|e|}} \cdot \frac{1 - C_V \dfrac{\mu n_m}{\Delta P_m} - C_f}{1 + C_s \dfrac{\Delta P_m}{e\mu n_m}} \tag{10}$$

where $\eta_P$, $\eta_m$ and $\eta_{Pm}$ represent the efficiency of the pump, the efficiency of the motor and the efficiency of the entire hydraulic system, respectively. $C_f$ represents the mechanical resistance coefficient; $C_V$ represents the resistance coefficient of laminar flow; $C_s$ represents the leakage coefficient of laminar flow; $\mu$ represents the hydraulic oil dynamic viscosity and $\Delta P_P$ and $\Delta P_m$ represent the oil pressure drop of pump and motor. According to the selected axial piston variable pump-quantitative motor model: $C_s = 0.8 \times 10^{-9}$; $C_V = 0.23 \times 10^{-6}$; $C_f = 0.01$; $\mu = 55.6 \times 10^{-3}$; $\Delta P_P = 8.6$ MPa and $\Delta P_m = 8.9$ MPa.

**Table 4.** Basic parameters of tractor.

| Tractor Model | Parameter |
|---|---|
| Maximum speed | 50 km/h |
| Engine rated power | 200 hp/149.1 kw |
| Engine rated speed | 2100 r/min |
| Maximum mass | 10,500 kg |
| Driving wheel tire | 480/80R50 |

The input speed was 2100 r/min, and the range of displacement ratio was [−1, 1]. The pump-motor hydraulic system test bench was built as shown in Figure 2. The simulation and test efficient curve of the pump-motor hydraulic system are shown in Figure 3. The average efficiency of the hydraulic system was 86.67%. Output gear efficiency was a combination of a planetary gear reducer ($n_{tg} = 0.985$) and differential gear ($n_{di} = 0.975$).

The coefficient of viscosity loss from the planetary gear mechanism to the output shaft gear mechanism is an empirical value, $\zeta = 0.1$ [Nm rad$^{-1}$ s]. The output efficiency can be expressed by Equation (11).

$$\eta_{out} = \left(\eta_{di}\eta_{tg}^{n_{tg}}\overline{\eta}_{pm} - \left|\frac{\xi\omega_{OUT}}{T_{OUT}}\right|\right) = \left(\eta_{di}\eta_{tg}^{n_{tg}}\overline{\eta}_{pm} - \left|\frac{\xi \times \omega_{DE} \times y_1}{T_{DE} \times y_2}\right|\right) \tag{11}$$

where $n_{tg}$ represents the number of transmission gear; $\overline{\eta}_{pm}$ represents the average efficiency of the hydraulic system and $\omega$ and $T$ represent the rotational speed and the torque of the shaft connecting the engine, respectively.

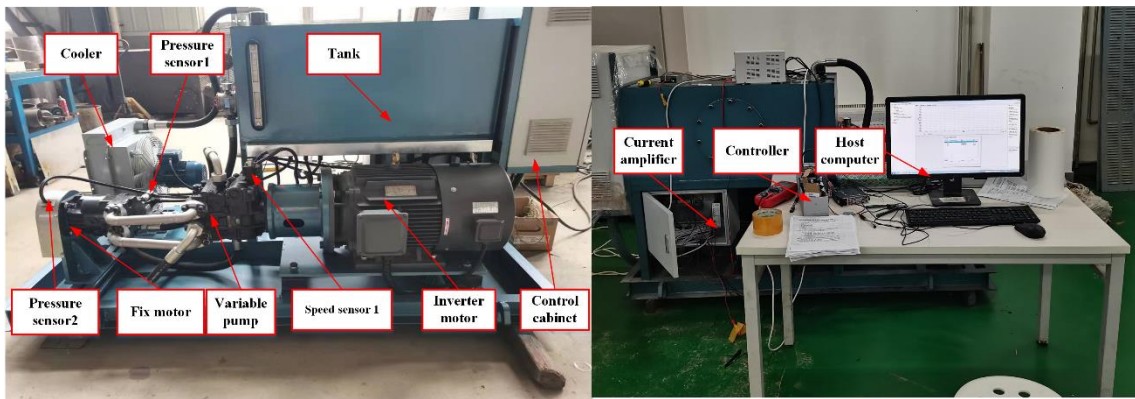

**Figure 2.** Pump-motor hydraulic system test bench.

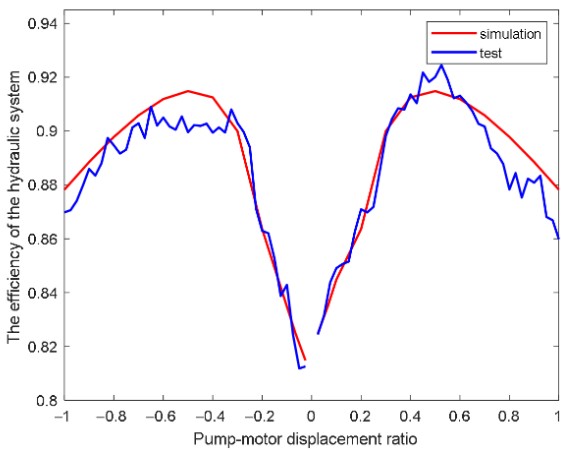

**Figure 3.** Pump-motor hydraulic system transmission efficiency diagram.

It can be seen from Equation (11) that the total efficiency is a function of $y_1$ and $y_2$, which are the functions of $k$. The layouts of *csr*, *src*, *scr* and *rsc* have beneficial efficiency in the effective range with the change in $k$. However, the efficiency ranges of the layouts *crs* and *rcs* vary greatly, and the efficiency is very low when the planetary gear parameter $k$ takes a more considerable role within the effective range (see Figure 4). Therefore, the four layouts of *csr*, *src*, *scr* and *rsc* are used as the selection criteria.

The above conclusion is that the layouts of *csr*, *crs*, *rcs* and *rsc* in the speed ratio are conducive to the transmission characteristics of the HMCVT. For efficiency optimisation, the layouts of *csr*, *src*, *scr* and *rsc* are beneficial to efficient work. In summary, the layout structures of *csr* and *rsc* are more suitable for the HMCVT in terms of transmission characteristics and efficiency.

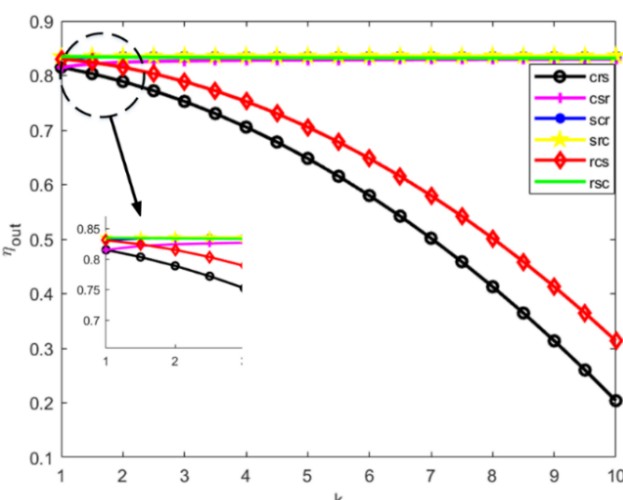

**Figure 4.** Efficiency graph of different layouts.

### 2.2. Design of Double Planetary HMCVT

According to the tractor parameters, the above conclusions and the transmission form of the existing high-horsepower tractor HMCVT, the design principle of the HMCVT is shown in Figure 5. In Figure 5, $i_x$ (x = 1, 2, . . . , 11) represents the gear pair, I II III IV V VI represent the transmission shaft, and yellow part represents pump and motor respectively. The hydraulic mechanical section is divided into four stages: HM1 (hydro-mechanical stage 1), HM2, HM3 and HM4. The power transmission routes in the four hydraulic mechanical sections are shown in Figure 6. It can be seen from the power flow route that the *csr* layout can be used in HM1 and HM3. The power was generated by the engine, one part of the power was transferred to the planet carrier of the planetary gear mechanism and the other part was transferred to the sun gear through the hydraulic transmission system. Finally, the two parts merged through the ring gear to the rear axle of the tractor. In the HM2 and HM4 stages, the structural layout of *rsc* was adopted. The power is yielded by the engine, one part of the power was transmitted to the ring gear of the planetary gear mechanism and the other part was transmitted to the sun gear of the planetary gear mechanism through the hydraulic transmission system. Finally, the two parts merged through the planet carrier to the tractor's rear axle.

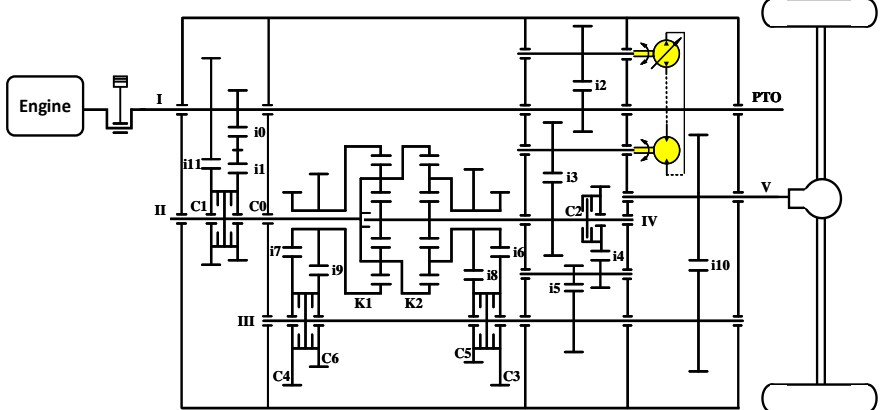

**Figure 5.** Schematic diagram of HMCVT transmission.

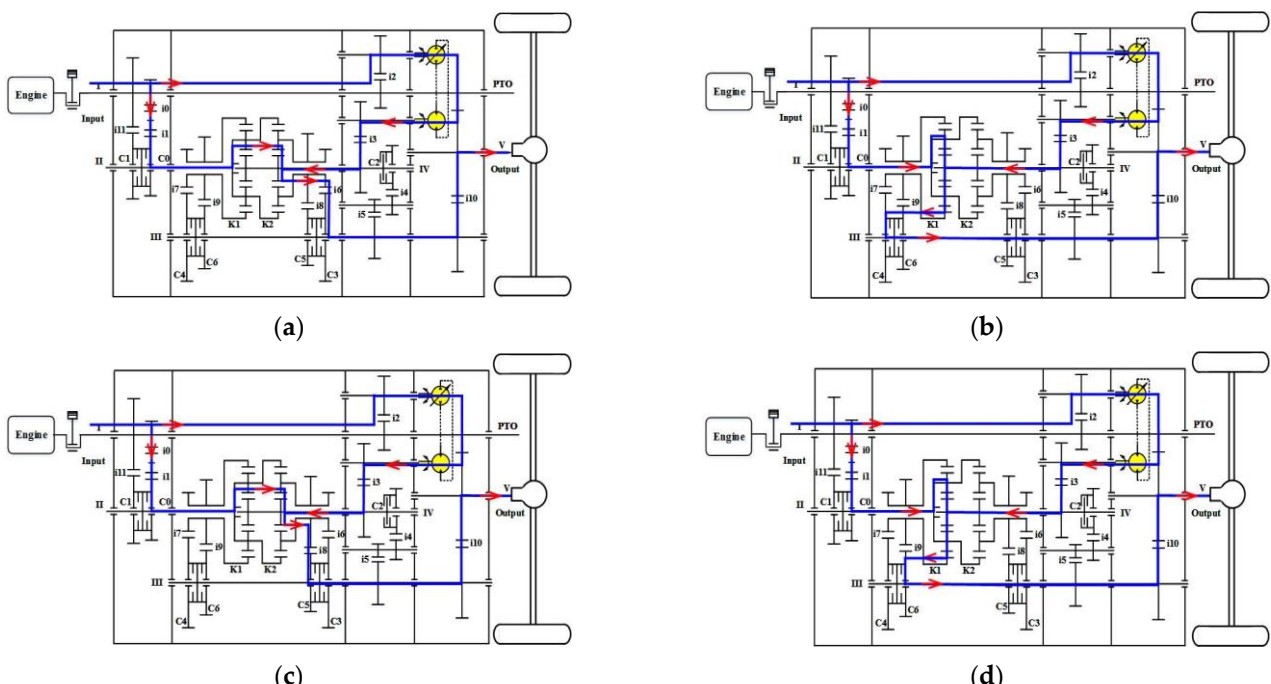

**Figure 6.** Power flow roadmap in HM*x*: (**a**) HM1; (**b**) HM2; (**c**) HM3 and (**d**) HM4.

The HMCVT speed regulation characteristics are shown in Figure 7. The clutches C3, C4, C5 and C6 are controlled to complete the switching of each section in HM1, HM2, HM3 and HM4, so the continuous speed change between stages is realized. The displacement ratio $e$ of the pump and the motor is controlled in order to achieve a continuous speed change in each stage.

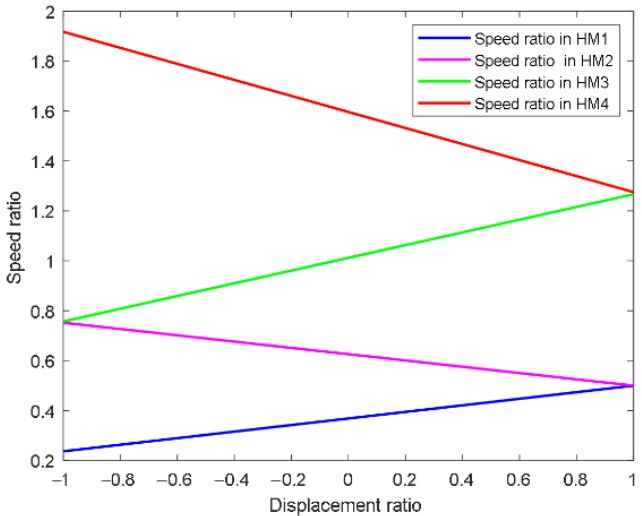

**Figure 7.** Speed regulation characteristics in hydro-mechanical stages of HMCVT.

The input coupled (IC) and output coupled (OC) are two primary forms of fixed-axis gear pair and planetary gear pair [17]. Because the output coupled (OC) has superior speed regulation characteristics, this study adopts the output coupled (OC) in the HMCVT. The speed ratio of each stage is shown in Figure 7 and Appendix A. Driven by a 200-horsepower engine, the HMCVT can achieve continuously variable speeds within the range of 0–50 km/h in the forward direction and 0–15 km/h in the reverse direction. In order to ensure the rationality of the tractor speed range distribution, the transmission scheme designed in this study adopts a multi-stage transmission type of equal ratio [18] which is

shown in Equations (12) and (13). According to the overall design requirements, the highest and lowest forward speeds in the hydro-mechanical stages of the tractor are 50 km/h and 4 km/h, respectively.

$$\tau = \sqrt[n]{\frac{v_{\max}}{v_{\min}}} \tag{12}$$

where $v_{\max}$ represents the highest vehicle speed in the hydraulic–mechanical stage; $v_{\min}$ represents the lowest vehicle speed in the hydro-mechanical stage and $n$ represents the number of hydro-mechanical stages.

$$\left.\begin{array}{c} \dfrac{1 + k_2 \dfrac{i_2 i_3}{i_0 i_1}}{-1 + k_2 \dfrac{i_2 i_3}{i_0 i_1}} = \tau \\[3ex] \dfrac{(1 + k_1) \dfrac{i_2 i_3}{i_0 i_1} + 1}{(1 + k_1) \dfrac{i_2 i_3}{i_0 i_1} - 1} = \tau \end{array}\right\} \quad \dfrac{i_2 i_3}{i_0 i_1} = \dfrac{\tau + 1}{(\tau - 1)k_2}, 1 + k_1 = k_2 \tag{13}$$

The transmission ratio of the hydro-mechanical stage is calculated to be 5.505 to 0.440 (Appendix B). The transmission ratio of each transmission gear pair and the value of $k$ are calculated as shown in Table 5.

**Table 5.** Gearbox gear pair transmission ratio and $k$ value.

| Parameter | Value | Parameter | Value |
|---|---|---|---|
| $i_0$ | 1.01 | $i_7$ | 1.70 |
| $i_1$ | 1.32 | $i_8$ | 0.54 |
| $i_2$ | 0.58 | $i_9$ | 0.48 |
| $i_3$ | 1.89 | $i_{10}$ | 1.20 |
| $i_4$ | 2.12 | $i_{11}$ | 1.33 |
| $i_5$ | 1.97 | $k_1$ | 2.96 |
| $i_6$ | 1.91 | $k_2$ | 3.96 |

## 3. Results

### 3.1. Analysis of Efficiency Characteristics for HMCVT

#### 3.1.1. Working Principle

The HMCVT is composed of four hybrid stages, HM1, HM2, HM3 and HM4, as well as pure hydraulic stage H and three reverse gears, RH, RHM1 and RHM1. The specific working process of the HMCVT is shown as follows:

(1)  At the starting stage of the tractor, the clutch C2 is engaged, clutches C0, C1, C3, C4, C5 and C6 are separated, and the HMCVT works in the hydrostatic transmission stage (H). When the tractor moves backwards, the clutches C0 and C1 are engaged, and the HMCVT works in the hydrostatic transmission of the reverse stage (RH). The power flow is shown in Figure 8.

(2)  When the tractor is working and transporting, the HMCVT will switch between the four stages of HM1, HM2, HM3 and HM4. The switching between the stages is completed by the clutches C1, C3, C4, C5 and C6. When the tractor backs up, the HMCVT is switched in two stages (RHM1 and RHM2), and the switching between the stages is realised by clutches C0, C3 and C4. The specific shift clutch control logic is shown in Table 6.

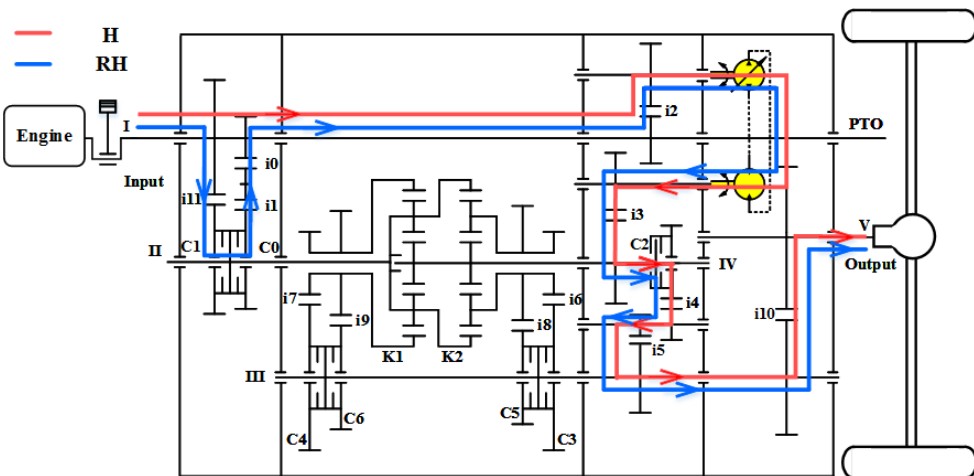

**Figure 8.** Power flow in pure hydraulic section.

**Table 6.** HMCVT shift clutch logic. (+: engage; − separate).

| Direction | Working Stage | C0 | C1 | C2 | C3 | C4 | C5 | C6 |
|-----------|---------------|----|----|----|----|----|----|----|
| Forward | H | − | − | + | − | − | − | − |
| | HM1 | − | + | − | + | − | − | − |
| | HM2 | − | + | − | − | + | − | − |
| | HM3 | − | + | − | − | − | + | − |
| | HM4 | − | + | − | − | − | − | + |
| Reverse | RH | + | + | + | − | − | − | − |
| | RHM1 | + | − | − | + | − | − | − |
| | RHM2 | + | − | − | − | + | − | − |

### 3.1.2. Theoretical Efficiency of the Hydro-Mechanical Stage

The efficiency loss of the entire HMCVT is divided into three parts: pump-motor hydraulic system, gear pair transmission and planetary mechanism transmission. The efficiency of mechanical transmission is mainly related to the lubrication of the shaft and the transmission efficiency of the gears [19]. The theoretical efficiency of mechanical transmission is mainly related to the transmission efficiency of shafts and gears. The transmission efficiency of each pair of gears is an empirical value of 0.985 [20]. According to the analysis in Figure 5, HM1 and HM3 are in *csr* layout, and HM2 and HM4 are in *rsc* layout. For this kind of structure, this paper computes the transmission efficiency of planetary using the engaging power method. The equivalent analyses of the planetary structure in each stage are shown in Figure 9.

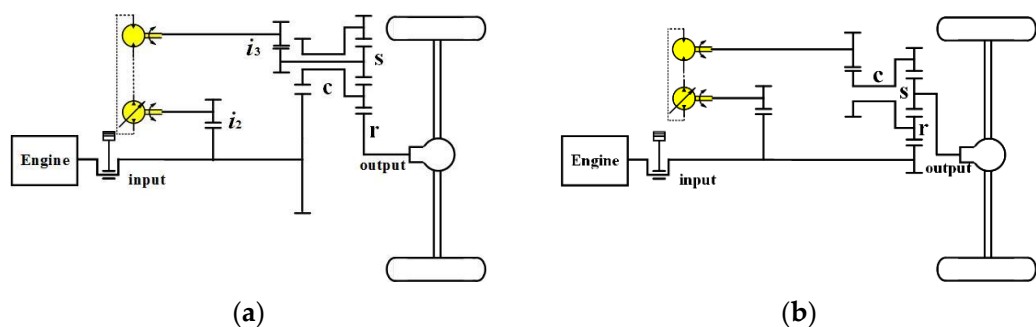

(**a**)             (**b**)

**Figure 9.** Equivalent transmission principle diagram: (**a**) HM1 (HM3) and (**b**) HM2 (HM4).

The design of this study is a closed planetary gear transmission system, and the meshing power method is usually used to calculate the transmission efficiency. The

efficiency of each section of HMCVT is analysed under the rated speed of 2100 r/min. In HM1, the HMCVT adopts a closed planetary gear structure, as shown in Figure 9a. The equivalent transmission ratio can be expressed as:

$$
\begin{cases}
i_{rI} = 1, i_{sI} = \dfrac{e}{i_2 i_3}, \\[2mm]
i^s_{cI} = i^s_{cr} i_{rI} = \dfrac{k_2}{1+k_2}, i^r_{cI} = i^r_{cs} i_{sI} = \dfrac{e}{(1+k_2) i_3 i_2}, i^r_{cs} = \dfrac{1}{1+k_2}, \\[2mm]
i^s_{cr} = \dfrac{k_2}{1+k_2}, i_{cI} = i^r_{cI} + i^s_{cI} = \dfrac{k_2}{1+k_2} + \dfrac{e}{(1+k_2) i_2 i_3}, i^r_{cI} i^s_{cI} = \dfrac{k_2 e}{(1+k_2)^2 i_2 i_3}
\end{cases}
\tag{14}
$$

where $s$, $r$ and $c$ represent the sun gear, ring gear and planet carrier, respectively; the subscripts $rI$, $sI$, $cI$, $cr$ and $cs$ represent the transmission ratios of the corresponding components of the equivalent transmission chain in Figure 9a, respectively, and the superscripts $r$ and $s$ represent the corresponding ring gear and sun gear components after simplification of the drive train in Figure 9a, respectively.

When $i^r_{cI} > 0$, $e > 0$, there is no circulating power in HM1. The transmission efficiency can be expressed as:

$$
\eta_{\mathrm{hm11}} = \{1 + |i_{Ic}| [i^s_{cI} - i^s_{cI} i_{cI}] \psi + |i^r_{cI}| (^1\big/_{\eta_{\mathrm{bI}}} - 1)]\}^{-1} \eta_{i10} \eta_{i11} \eta_{i3}
\tag{15}
$$

$$
\eta_{\mathrm{hm11}} = \{1 + \left| \dfrac{1+k_2}{k_2 + \frac{e}{i_2 i_3}} \right| [\left| \dfrac{k_2}{1+k_2} \left( \dfrac{1}{1+k_2} - \dfrac{e}{(1+k_2) i_2 i_3} \right) \right| \psi + \left| \dfrac{e}{(1+k_2) i_2 i_3} \right| (^1\big/_{\overline{\eta}_{\mathrm{pm}} \eta_{i2} \eta_{i3}} - 1)]\}^{-1} \eta_{i10} \eta_{i11} \eta_{i6}
\tag{16}
$$

When $i^r_{cI} < 0$, $e < 0$, there is circulating power in HM1. The transmission efficiency can be expressed as:

$$
\eta_{\mathrm{hm12}} = \{1 + |i_{Ic}| [i^s_{cI} - i^s_{cI} i_{cI}] \psi + |i^r_{cI}| (1 - \eta_{\mathrm{bI}})]\}^{-1} \eta_{i10} \eta_{i11} \eta_{i5}
\tag{17}
$$

$$
\eta_{\mathrm{hm12}} =
\{1 + \left| \dfrac{1+k_2}{k_2 + \frac{e}{i_2 i_3}} \right| [\left| \dfrac{k_2}{1+k_2} \left( \dfrac{1}{1+k_2} - \dfrac{e}{(1+k_2) i_2 i_3} \right) \right| \psi + \left| \dfrac{e}{(1+k_2) i_2 i_3} \right| (1 - \overline{\eta}_{\mathrm{pm}} \eta_{i2} \eta_3)]\}^{-1} \eta_{i10} \eta_{i11} \eta_{i6}
\tag{18}
$$

where $\eta_{\mathrm{hm11}}$ represents transmission efficiency when $e > 0$; $\eta_{\mathrm{hm12}}$ represents the transmission efficiency when $e < 0$; $k_2$ represents the structure parameter of the second planet carrier; $i$ represents the transmission ratio of the gear pair; $\psi$ represents the equivalent transmission loss coefficient; $\eta$ represents transmission efficiency; the subscripts 1, 2, 3, … , 11 represent the corresponding gear in Figure 5, respectively, and the subscript pm represents the pump-motor hydraulic system.

In HM2, the HMCVT adopts a closed planetary gear structure, as shown in Figure 9b. The equivalent transmission ratio can be expressed as:

$$
\begin{cases}
i_{sI} = \frac{e}{i_2 i_3}, i_{cI} = 1, i^s_{rI} = i^s_{rc} i_{cI} = \frac{1+k_1}{k_1} \\[2mm]
i^c_{rI} = i^c_{rs} i_{sI} = -\frac{e}{k_1 i_2 i_3}, i_{Ir} = 1/i_{rI}, i^s_{rc} = \frac{1+k_1}{k_1}, \\[2mm]
i^c_{rs} = -\frac{1}{k_1}, i_{rI} = i^s_{rI} + i^c_{rI} = \frac{1+k_1}{k_1} - \frac{k_1 e}{i_2 i_3}, i^s_{rI} i^c_{rI} = -\frac{(1+k_1) e}{k^2_1 i_2 i_3}
\end{cases}
\tag{19}
$$

When $i^s_{rI} < 0$, $e < 0$, there is circulating power in HM2. The transmission efficiency can be expressed as:

$$
\eta_{\mathrm{hm21}} = \{1 + |i_{Ir}| [i_{rI} - i_{cI}] \psi + |i^c_{rI}| (1 - \eta_{\mathrm{bI}})]\}^{-1} \eta_{i10} \eta_{i11} \eta_{i4}
\tag{20}
$$

$$\eta_{hm21} = \{1 + \left|\frac{k_1}{1 + k_1 - \frac{e}{i_2 i_3}}\right| \left[\left|\frac{1}{k_1}(1 - \frac{e}{i_2 i_3})\right|\psi + \left|\frac{e}{k_1 i_2 i_3}\right|(1 - \bar{\eta}_{pm}\eta_{i2}\eta_{i3})\right]\}^{-1} \eta_{i10}\eta_{i11}\eta_{i7} \tag{21}$$

When $i_{rI}^s > 0$, $e < 0$, there is no circulating power in HM2. The transmission efficiency can be expressed as:

$$\eta_{hm22} = \{1 + |i_{Ir}|[i_{rI} - i_{cI}]\psi + |i_{rI}^c|(\left.{}^{1}\middle/{}_{\eta_{bI}}\right. - 1))\}^{-1}\eta_{i10}\eta_{i11}\eta_{i4} \tag{22}$$

$$\eta_{hm22} = \{1 + \left|\frac{k_1}{1 + k_1 - \frac{e}{i_2 i_3}}\right| \left[\left|\frac{1}{k_1}(1 - \frac{e}{i_2 i_3})\right|\psi + \left|\frac{e}{k_1 i_2 i_3}\right|(\left.{}^{1}\middle/{}_{\bar{\eta}_{pm}\eta_{i2}\eta_{i3}}\right. - 1))\right]\}^{-1} \eta_{i10}\eta_{i11}\eta_{i7} \tag{23}$$

Similarly, the efficiency of HM3 and HM4 can be expressed as:

$$\eta_{hm31} = \{1 + \left|\frac{1 + k_2}{k_2 + \frac{e}{i_2 i_3}}\right| \left[\left|\frac{k_2}{1 + k_2}(\frac{1}{1 + k_2} - \frac{e}{(1 + k_2)i_2 i_3})\right|\psi + \left|\frac{e}{(1 + k_2)i_2 i_3}\right|(\left.{}^{1}\middle/{}_{\bar{\eta}_{pm}\eta_{i2}\eta_{i3}}\right. - 1))\right]\}^{-1} \eta_{i10}\eta_{i11}\eta_{i8}, e \geq 0 \tag{24}$$

$\eta_{hm32} =$
$$\{1 + \left|\frac{1 + k_2}{k_2 + \frac{e}{i_2 i_3}}\right| \left[\left|\frac{k_2}{1 + k_2}(\frac{1}{1 + k_2} - \frac{e}{(1 + k_2)i_2 i_3})\right|\psi + \left|\frac{e}{(1 + k_2)i_2 i_3}\right|(1 - \bar{\eta}_{pm}\eta_{i2}\eta_{i3})\right]\}^{-1} \eta_{i10}\eta_{i11}\eta_{i8}, e < 0 \tag{25}$$

$$\eta_{hm41} = \{1 + \left|\frac{k_1}{1 + k_1 - \frac{e}{i_2 i_3}}\right| \left[\left|\frac{1}{k_1}(1 - \frac{e}{i_2 i_3})\right|\psi + \left|\frac{e}{k_1 i_2 i_3}\right|(1 - \bar{\eta}_{pm}\eta_{i2}\eta_{i3})\right]\}^{-1} \eta_{i10}\eta_{i11}\eta_{i9}, e \geq 0 \tag{26}$$

$$\eta_{hm42} = \{1 + \left|\frac{k_1}{1 + k_1 - \frac{e}{i_2 i_3}}\right| \left[\left|\frac{1}{k_1}(1 - \frac{e}{i_2 i_3})\right|\psi + \left|\frac{e}{k_1 i_2 i_3}\right|(\left.{}^{1}\middle/{}_{\bar{\eta}_{pm}\eta_{i2}\eta_{i3}}\right. - 1))\right]\}^{-1} \eta_{i10}\eta_{i11}\eta_{i9}, e < 0 \tag{27}$$

where the subscripts $\eta_{hm11}$ and $\eta_{hm12}$ represent the efficiencies of $e \geq 0$ and $e < 0$ in HM1, respectively; similarly, $\eta_{hm21}$, $\eta_{hm22}$, $\eta_{hm31}$, $\eta_{hm32}$, $\eta_{hm41}$ and $\eta_{hm42}$ represent the efficiencies of $e \geq 0$ and $e < 0$ in corresponding stages, respectively; $k_1$ and $k_2$ represent the corresponding planetary gear characteristic parameters; $e$ represents the pump-motor displacement ratio; the subscripts $\eta_{i0}, \eta_{i1}, \ldots, \eta_{i11}$ and $\eta_{pm}$ represent the efficiencies of the corresponding gears and the efficiency of the pump-motor hydraulic system, respectively, and $i$ represents gear ratio.

From Equations (21), (23), (26) and (27), it can be seen that the efficiency of the HMCVT in HM1 and in HM3 are the same, except for the transmission efficiency of the last gear. The efficiency of the gear pair in this study is the same value. Different clutches will not affect the efficiency characteristics of the transmission. Therefore, the efficiency of the HMCVT with the change of the displacement ratio is the same in HM1 and HM3, as shown in Figure 10a. Similarly, the efficiency of HMCVT is the same in HM2 and HM4, as shown in Figure 10b.

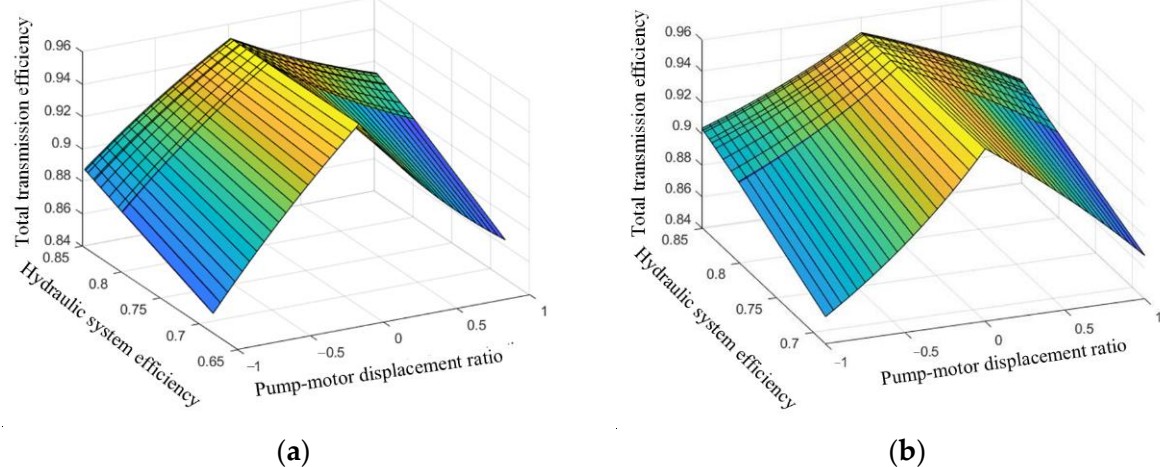

(**a**)                                          (**b**)

**Figure 10.** The efficiency of the hydro-mechanical stage: (**a**) Efficiency graph of HM1 (HM3) and (**b**) efficiency graph of HM2 (HM4).

### 3.1.3. Power Split Characteristics

The hydraulic power split ratio (Equation (28)) is the ratio of the hydraulic system drive branch to the output power of the HMCVT [21]. The power split characteristic reflects the relationship between the hydraulic power split ratio of the HMCVT and the speed of the vehicle [22].

$$\lambda = P_m / P_{out} \tag{28}$$

$$P_m = T_m \omega_m = T_m \omega_{in} \frac{e}{i_2} \tag{29}$$

where $P_m$, $T_m$ and $\omega_m$ represent the output power, torque and speed of the hydraulic system, respectively; $\omega_{in}$ represents the input speed of the hydraulic system; $i_n$ represents the transmission ratio of the corresponding $n$ gear; $P_{out}$ represents the output power of HMCVT and $\lambda$ represents the hydraulic power split ratio.

The output torque and the output power of HMCVT in HM1 are expressed as:

$$T_{out} = (1 + k_2)i_3 i_6 i_{10} T_m \tag{30}$$

$$P_{out} = T_{out}\omega_{out} = (1 + k_2)i_3 i_6 i_{10} T_m \frac{1}{(1 + k_2)i_2 i_3}(e + k_2 \frac{i_2 i_3}{i_0 i_1})\frac{1}{i_6 i_{10}}\omega_{in} = \frac{T_m \omega_{in}}{i_2}(e + k_2 \frac{i_2 i_3}{i_0 i_1}) \tag{31}$$

Thus, the hydraulic power split ratios of HM1, HM2, HM3 and HM4 are expressed as:

$$\lambda_1 = \lambda_3 = \frac{P_m}{P_{out}} = \frac{e}{e + k_2 \frac{i_2 i_3}{i_0 i_1}} \tag{32}$$

$$\lambda_2 = \lambda_4 = \frac{e}{e - (1 + k_1)\frac{i_2 i_3}{i_0 i_1}} \tag{33}$$

where $\lambda_i$ represents the hydraulic power split ratio of HM$i$; $k$ represents characteristic parameters of the planetary row structure and $e$ represents the displacement ratio of pump-motor hydraulic system.

The variation of the hydraulic power split ratio in the four stages with the vehicle speed is shown in Figure 11. The hydraulic power split ratio in HM1 and HM3 improves with the

increase in vehicle speed, because the derivative to $e$ is greater than zero (Equation (34)) when $e \in (-1, 1)$.

$$\frac{d\lambda_1}{de} = \frac{k_2 \frac{i_2 i_3}{i_0 i_1}}{\left(e + k_2 \frac{i_2 i_3}{i_0 i_1}\right)^2} > 0, e \in (-1, 1) \tag{34}$$

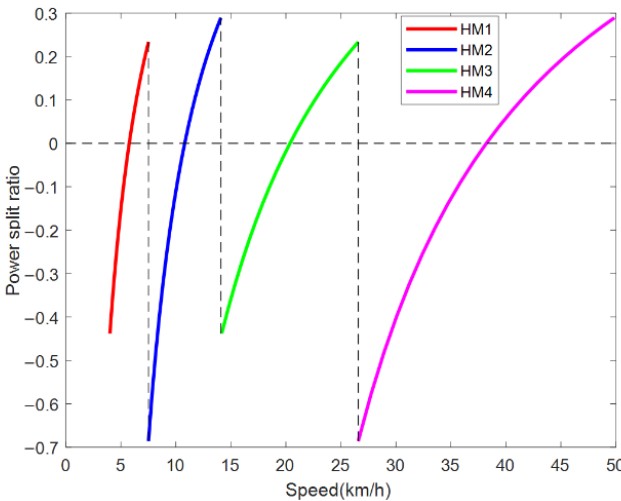

**Figure 11.** Power split ratio changes with vehicle speed.

The hydraulic power split ratio decreases with the increase in vehicle speed at HM1 and HM3, and when the displacement $e$ is 0, the hydraulic power split ratio is 0, which corresponds to the highest efficiency point.

It can be seen from Figure 11 that the displacement ratio $e$ of the two stages of HM2 and HM4 ranges from 1 to $-1$. The derivative of $e$ is less than 0 (Equation (35)), so the power split ratio declines when $e \in (-1, 1)$. However, when the speed increases ($e \in (1, -1)$) in HM2, the hydraulic power split ratio will increase gradually.

$$\frac{d\lambda_2}{de} = \frac{-(1 + k_1) \frac{i_2 i_3}{i_0 i_1}}{\left((1 + k_1) \frac{i_2 i_3}{i_0 i_1} - e\right)^2} < 0, e \in (-1, 1) \tag{35}$$

It can be seen from Figure 11 that the hydraulic power split ratio is different at the switching point. When switching, the absolute value of the splitting ratio in HM1 and HM3 is smaller than that in HM2 and HM4. Additionally, splitting ratios in HM2 and HM4 have negative values, which shows the circulating power. Therefore, the efficiency of HM1 and HM3 at the switching point is higher than that of HM2 and HM4. When the speed is 4–6 km/h, 7.5–10.8 km/h, 14.2–20.4 km/h and 26.6–38.2 km/h, the hydraulic power split ratio has a negative value. Thereby, the circulating power is generated in the above stages to reduce the transmission efficiency of HMCVT. However, when the speed is 6–7.5 km/h, 10.8–14.2 km/h, 20.4–26.6 km/h and 38.2–50 km/h, the hydraulic power split ratio has a positive value, which reveals no circulating power. Therefore, the shifting point should be selected in the low-ranking gear area to ensure the higher shifting efficiency of the HMCVT.

### 3.2. Efficiency Characteristics under Maximum Load

Differently from the traditional method of calculating the efficiency of the HMCVT [23], the HMCVT and the engine are regarded as an entire system in this article. Thus, the efficiency of the HMCVT is limited by the extreme transmission characteristics of the engine. In order to evaluate the limit of the HMCVT's efficiency, the limit torque value corresponding to the external characteristic curve of the engine is loaded in order to obtain efficiency of the maximum load. According to the above mathematical model, the hardware-

in-the-loop test bench of the HMCVT is established as shown in Figure 12. The hardware portion of the HIL mainly includes the host computer, the pump-motor hydraulic system test bench, the clutch system test bench, dSPACE, the host computer, the controller, etc. The pump-motor hydraulic system parameters are shown in Appendix C. The HIL hardware parameters are shown in Appendix D, and the clutch system parameters are shown in Appendix E. First, the efficiency characteristic under the external characteristic (maximum load) is calculated.

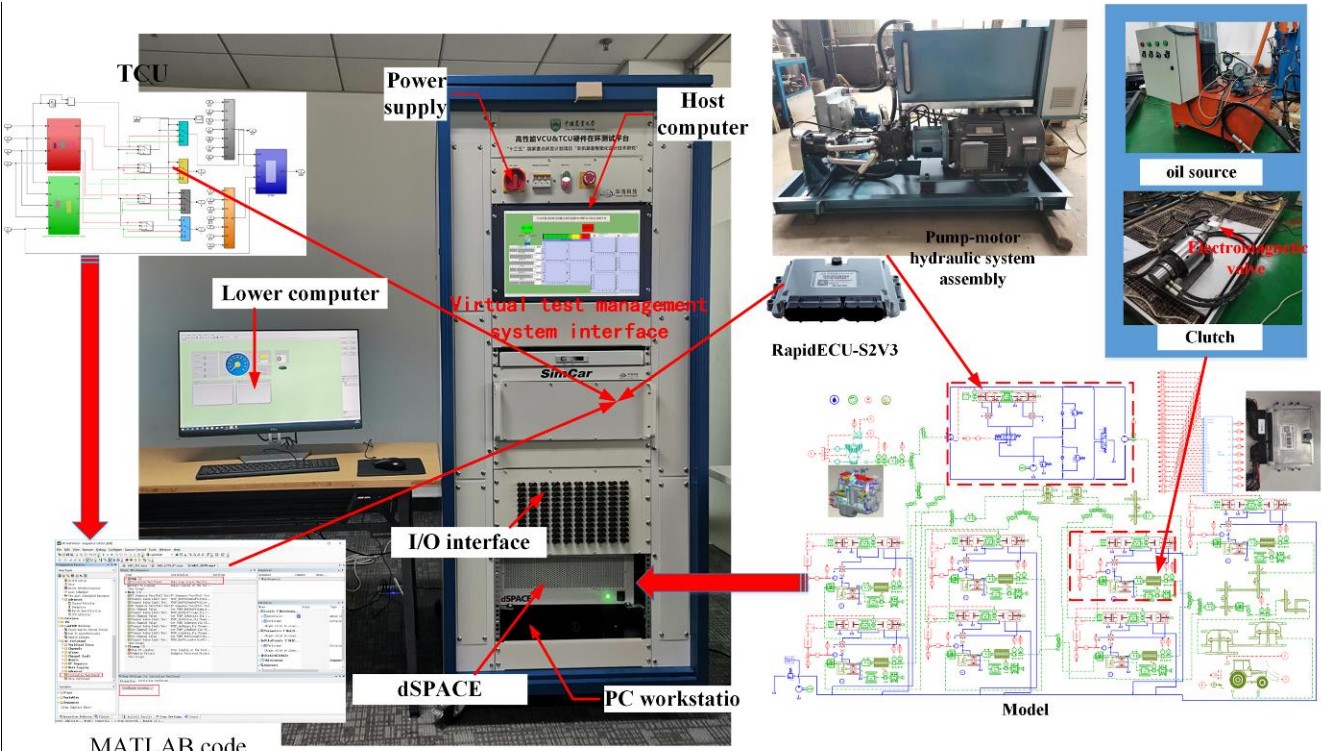

**Figure 12.** HMCVT hardware in the loop test platform.

It can be seen from Figure 13 that the efficiency of each stage increases first and then decreases, which is consistent with the trend of variation. The highest efficiency point in each stage is obtained when the vehicle speed is 5.76 km/h, 10.8 km/h, 20 km/h and 38.2 km/h, corresponding to the full mechanical point. The average efficiency of HM3 is higher under the external characteristic limit of torque, so this stage should be selected as much as possible in a high-power operation. From Figures 11 and 13, it can be seen that the hydraulic power split ratio of the transmission part with circulating power has a negative value, and the average efficiency is generally lower than the average efficiency when there is no circulating power. For example, when $e \in (-1, 0)$ in HM1, the power split ratio has a negative value, and there is circulating power. Thus, the average efficiency of $e \in (-1, 0)$ in HM1 is lower than the average efficiency when $e \in (0, 1)$. The average efficiency of HM4 is lower than that of HM3. Because HM4 is a high-speed stage, the high-speed rotation of the pump-motor, gear and shaft consume energy and bring heat. In this study, the HMCVT is designed on the basis of efficiency characteristics, so the lowest efficiency is above 80%.

### 3.3. Efficiency Characteristics under Variable Load

The tractors have different requirements for driving speed in different operating conditions [24]. A given driving speed can be achieved by adjusting the engine speed and the transmission ratio of the gearbox [25]. When the load changes, in order to maintain a given driving speed, the engine input speed torque and gearbox transmission ratio need to be adjusted. The gearbox will have different transmission efficiencies for different engine output torques and output speeds at the same driving speed. Therefore, different output

efficiencies will vary with the load under the same operating speed, and the efficiency characteristic field of HMCVT under variable load can be obtained.

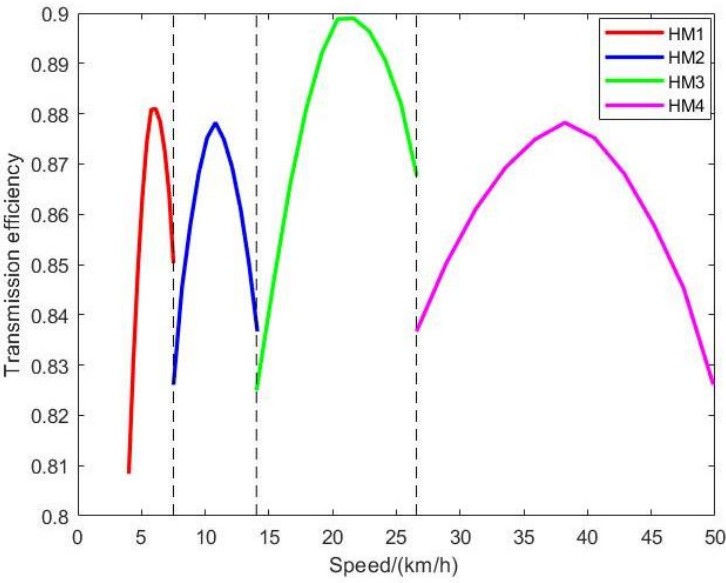

**Figure 13.** The efficiency of hydraulic–mechanical stages under full load.

The process of obtaining the efficiency characteristic field of the transmission system under variable load is:

(1)　The driving speed is obtained according to the operating conditions;

(2)　The engine output speed is calculated by solving different displacement ratios;

(3)　The output torque of the engine is adjusted when the engine has different output speeds, so that the output torque of the HMCVT reaches the set value of the variable load;

(4)　The different output speeds of the engine are obtained at a specific speed, according to the model. In addition, the engine torque corresponding to different loads and the transmission efficiency of the corresponding gearbox is calculated;

(5)　The universal characteristic area, which is defined by the external characteristics of the engine (red line in Figure 14), is gridded to derive the transmission efficiency characteristic field at a certain tractor speed;

(6)　The acquired data are subjected to a multiple regression analysis to obtain the characteristic efficiency field at a certain speed.

In order to clearly show the influence of the independent variable of the engine on the transmission efficiency field when the load is changed, the transmission efficiency of HM1 to HM2 from 4 km/h to 11 km/h is analysed as shown in Figures 14–16. The variation in the efficiency of the shift point needs to be specifically analysed, so the adjacent speed (7 km/h, 8 km/h) of the shift point is calculated. As the engine speed increases, the gearbox speed ratio will decrease to maintain the same driving speed. When the speed ratio decreases, the displacement ratio will approach 0 and then increase in the opposite direction. The hydraulic power split ratio also appears first to decrease and then to increase. Therefore, the efficiency of the HMCVT at a certain speed of each stage increases first and then decreases with the rotation speed increase, as shown in Figure 14.

Figure 14a–c show the changes in the efficiency of the HM1 when the vehicle speed is 4 km/h, 6 km/h and 7 km/h with different engine speeds and torque, which are yielded with different loads. At the same speed, the efficiency increases first and then decreases with the improvement of engine speed and torque. In order to maintain the stability of the speed, the displacement ratio will first increase and then decrease, so the efficiency trend is the same as that shown in Figure 13. However, when the torque exceeds the

limit, the energy consumption increases and the efficiency decreases. With the increase in vehicle speed, the highest efficiency zone moves to the right and the highest efficiency value decreases. When the speed ratio is a fixed value, a higher engine output speed is required to match the high vehicle speed. Moreover, depending on the viscous damping, the increase in engine speed causes the HMCVT to lose more energy.

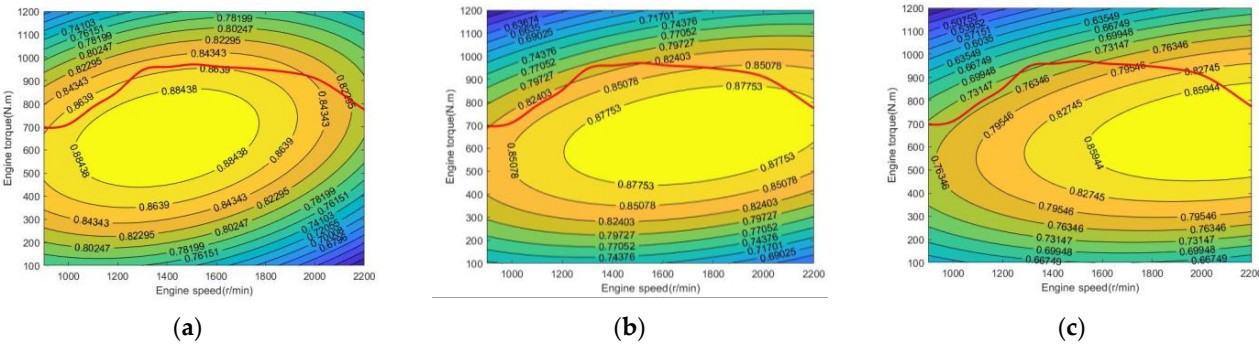

(**a**)        (**b**)        (**c**)

**Figure 14.** Efficiency characteristic field of HM1 at different speeds: (**a**) 4 km/h, (**b**) 6 km/h and (**c**) 7 km/h.

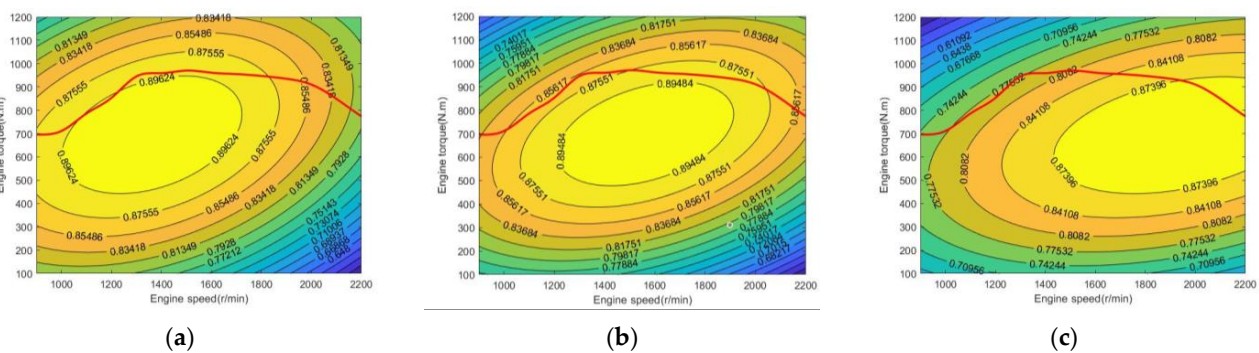

(**a**)        (**b**)        (**c**)

**Figure 15.** Efficiency characteristic field of HM2 at different speeds: (**a**) 8 km/h, (**b**) 9 km/h and (**c**) 11 km/h.

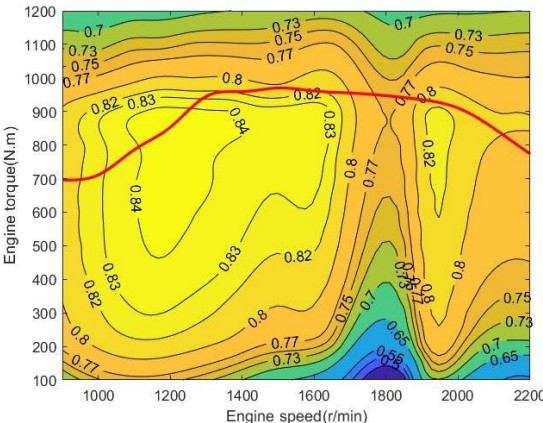

**Figure 16.** The characteristic efficiency field of HMCVT at the shift point (7.5 km/h) from HM1 to HM2.

Figure 15a–c show that the efficiency of the HMCVT varies with engine speed and torque when the load changes in HM2.

It can be seen from Figure 15 that the change in the highest efficiency area is similar to that of HM1, but the highest efficiency performance in HM2 has hardly changed. Normally, the higher engine torque is required for the larger variable load torque, which contributes

to higher transmission efficiency. With the increase in the load power, the ratio of internal power consumption to total power gradually decreases, which improves the overall efficiency of the HMCVT. Nevertheless, when the engine torque (or gearbox load torque) is too high, the transmission efficiency of the gearbox is affected by the volumetric efficiency of the pump-motor hydraulic system, and will decrease as the torque increases.

Figure 16 shows the efficiency field at the shift point from HM1 to HM2. It can be seen that the efficiency is relatively low, and the different transmission modes between the two sections at the shift point cause fluctuations in the efficiency curve. In Figure 16, there are two efficiency peaks, one when the transmission efficiency is 1200 r/min and one when it is 1900 r/min. The efficiency peak at 1900 r/min is smaller than the efficiency peak at 1200 r/min. This is because the absolute value of the power split ratio in HM2 is greater than the power split ratio of HM1 at the switching point, and the circulating power exists in the HM2 shift area. The maximum efficiency fluctuation is at the speed of 1800 r/min, which is caused by the impact of shift jerk.

## 4. Conclusions

(1)  The study proposes a mathematical method to define the layouts of the single row planetary HMCVT. This method uses the planetary gear characteristic parameter $k$ as the independent variable to combine the three parts of the planetary gear with the mechanical system, which can utilize different layouts. According to the transmission ratio characteristics and efficiency characteristics of different layouts, the suitable layouts are selected to design a double planetary HMCVT, which is designed with optimal transmission efficiency and gear ratio range.

(2)  Based on the characteristics of the hydraulic power split ratio, the mathematical model, the pump motor hydraulic test bench and the clutch system test bench, the hardware-in-the-loop system of HMCVT is designed, and the engine-HMCVT-load is built as an efficient whole.

(3)  The efficiency characteristic field in the maximum load and the variable load (5–11 km/h) are designed and analysed, respectively. The results show that the efficiency characteristics of HM2 and HM3 are better in the full load, so the speed range should be controlled at this stage as much as possible during operation. The transmission efficiency of HM2 with a power split is lower than that of HM1 at the shift point, so the shift is generally at a low gear.

(4)  This study completed the design portion and the characteristic analysis portion. However, the hardware in the loop test of the HMCVT is the first step. In the future, a control algorithm needs to be added to control the TCU of HMCVT, in order to test the reliability of HMCVT shifting. Then, further real vehicle tests are required to verify the designed HMCVT.

**Author Contributions:** Conceptualization, Z.Z. (Zhongxiang Zhu); methodology, J.L.; software, J.L., Z.Z. (Zhiqiang Zhai) and L.W.; formal analysis, B.H. and Y.D.; funding acquisition, Z.Z. (Zhongxiang Zhu) and Y.Z. All authors have read and agreed to the published version of the manuscript.

**Funding:** This research was funded by National Key Research and Development Program of China, grant number 2020YFB1713502; National Natural Science Foundation of China, grant number 52072407; National Natural Science Foundation of China, grant number 52175259.

**Institutional Review Board Statement:** Not applicable.

**Informed Consent Statement:** Not applicable.

**Data Availability Statement:** Not applicable.

**Conflicts of Interest:** The authors declare no conflict of interest.

## Appendix A. Speed Ratio of Each Gear

| Stage | Speed Ratio | Displacement Ratio $e$ |
|---|---|---|
| H | $i_h = -\dfrac{e}{i_2 i_3 i_4 i_5 i_{10}}$ | $0 \sim -1$ |
| HM1 | $i_{HM1} = \dfrac{1}{(1+k_2)i_2 i_3}\left(e + k_2 \dfrac{i_2 i_3}{i_0 i_1}\right)\dfrac{1}{i_6 i_{10}}$ | $-1 \sim +1$ |
| HM2 | $i_{HM2} = \dfrac{1}{k_1 i_2 i_3}\left[-e + (1+k_1)\dfrac{i_2 i_3}{i_0 i_1}\right]\dfrac{1}{i_7 i_{10}}$ | $+1 \sim -1$ |
| HM3 | $i_{HM3} = \dfrac{1}{(1+k_2)i_2 i_3}\left(e + k_2 \dfrac{i_2 i_3}{i_0 i_1}\right)\dfrac{1}{i_8 i_{10}}$ | $-1 \sim +1$ |
| HM4 | $i_{HM4} = \dfrac{1}{k_1 i_2 i_3}\left[-e + (1+k_1)\dfrac{i_2 i_3}{i_0 i_1}\right]\dfrac{1}{i_9 i_{10}}$ | $+1 \sim -1$ |

## Appendix B. Transmission Ratio Range and Output Speed Range of the HM stages

| Stage | Transmission Ratio Range | Output Rotation Speed Range (r/min) |
|---|---|---|
| HM1 | 5.505~2.928 | 381.47~717.21 |
| HM2 | 2.928~1.558 | 717.21~1347.88 |
| HM3 | 1.558~0.828 | 1347.88~2536.23 |
| HM4 | 0.828~0.440 | 2536.23~4772.73 |

## Appendix C. Selection of Key Components of Pump-Motor Hydraulic System

| Name | Model Parameters | Function |
|---|---|---|
| Variable pump | T90-R-075 | Power input, adjustable displacement |
| Quantitative motor | 90-M-075-W-EBA | Power output |
| Proportional control valve | PJ-440 | Control displacement ratio |
| Charge pump | NB-20 | Supplement hydraulic oil, prevent leakage |
| Inverter motor | YVF-180MB35-A | Power source |
| Speed sensor | NCJ-03 | Speed measurement and real-time feedback |
| Work pressure | $\geq 5$ MPa | System pressure minimum |
| inverter | SZTU-1100YR-B | Control motor speed |
| Cooler | T4-5204-203 | Cooling down the hydraulic system |

## Appendix D. HIL Hardware Selection and Technical Parameters

| Device Name | Specifications | Parameter |
|---|---|---|
| Host computer | Intel Core i7-10700 | Windows 10, 2.90 GHz, 16.0 GB |
| Processor board | DS1007 | QorlQ P5020, 2.0 GHz |
| TCU | RapidECU-S2V3 | MPC5674F, 32 bit, F160 MHz |
| CAN board | DS4302 | 4 channels, 16 bit, 1 Mbps |
| USB-CAN adapter | USB-CAN-2E-U | 2 |

## Appendix E. Clutch Parameter Table

| Parameter Name | Parameter Value |
|---|---|
| Pre-set pressure | 2 MPa |
| Transmit torque | 1637.9 N·m |
| Reserve factor | 2.795 |
| Clutch piston outer radius | 68.5 mm |
| Clutch piston inner radius | 30 mm |
| Piston stroke | 2 mm |
| Spring rate | 16.7 N/mm |
| Number of springs | 15 per side |
| Inlet diameter | 5 mm |
| Effective radius of friction plate and lining | 124 mm |
| Number of friction plates and linings | 5 |
| Solenoid valve working pressure | 2 MPa |
| Workflow | 15~100 L/min |
| Supply voltage | 0~24 V |
| Pressure sensor | MIK-P300 |

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
