# Peer review of "Research on Design and Analysis Method of the Double Planetary HMCVT Based on High Efficiency Transmission"

_agriculture, doi:10.3390/agriculture12111958_

Round 1

Reviewer 1 Report

The paper deals with the efficiency estimation in a Hydro-Mechanical CVT. The paper is of interest and the topic is surely trendsetting, but the presentation and the language make comprehension very difficult. In the reviewer's opinion, for improvement of publication, a deep reorganization of the paper and a deep review of the language is mandatory.

In addition, it is not clear how the experimental results on the hydraulic transmission are used in the computation of the overall efficiency.

Figure 9 seems to report the same scheme, please check.

Author Response

Responses to Comments and Suggestions

Dear reviewer,

Thank you for your comments concerning our manuscript. The comments are all valuable and very helpful for revising and improving our paper, as well as the important guiding significance to our research. We have studied comments carefully and have made correction which we hope to meet with your approval.

We have made changes for the manuscript, as shown in the updated version. In the updated version, the changes are marked up using the “Track Changes” function (according to the editor's request). The following are reply to the reviewer:

Main comments:

Comment 1: The paper deals with the efficiency estimation in a Hydro-Mechanical CVT. The paper is of interest and the topic is surely trendsetting, but the presentation and the language make comprehension very difficult. In the reviewer's opinion, for improvement of publication, a deep reorganization of the paper and a deep review of the language is mandatory.

Response 1: Thanks for the expert’s advice. The language of the manuscript is revised based on comment of reviewer and the manuscript is further reorganized. For example, the grammar in the text has been checked and modified; the layout definition of single planet row is described in detail again; the conclusion is revised and simplified, and the future work is proposed.

Comment 2: In addition, it is not clear how the experimental results on the hydraulic transmission are used in the computation of the overall efficiency.

Response 2: Thanks for the expert’s advice. The transmission efficiency of the pump-motor hydraulic transmission system was measured through the pump-motor test bench (Figure. 2), which was compared with the simulation value (Equation (10)) at different displacement, as shown in Figure. 3. Then the average transmission efficiency  was calculated through the test values under different displacement and used in equation 11 to obtain the total transmission efficiency

Comment 3: Figure 9 seems to report the same scheme, please check.

Response 3: Thanks for the expert’s advice. We are sorry for making the mistakes. The Figure 9 has been modified according to the engaging power method of the equivalent gear.(see word)

The above are our team's answers to the experts' questions. Once again, we express our respect for the expert. We very much hope that these explanations will enable the expert and readers to further understand our research work.

    We tried our best to improve the manuscript and made some changes in the manuscript.

    Once again, we appreciate for reviewers’ warm work earnestly, and hope that the revised version will meet with approval.

Thank you and best regards.

Reviewer 2 Report

The work presented for review is an interesting presentation of the process of designing and analyzing the Double Plan- 2 HMCVT Based on High Efficiency Transmission.

I highly appreciate the presented work, I would suggest that you consider dividing the work into smaller, separated fragments in which the goal that the authors wanted to achieve was precisely defined.

In the presented such extensive study, when reading, the main idea of ​​the work is lost due to the many threads touched upon in it.

The chapter on methodology is significantly expanded in the work, and some methodological elements are included in the chapter results. Although this is consistent with the logic of the presented study, it makes it difficult to separate the methodological part (description of measurement stations, course of tests, description of the calculation methods used) from the results in which the obtained results should be found.

Detailed comments are provided below,

1. Line 50. It's kw should be kw

2. Line from 149 to 154 there is a description of scr, and below there are no descriptions of other solutions, ie: src, rsc, rcs crs csr or a comment regarding the lack of such descriptions.

3. Figure 1. There is no description of the vertical bar on the right side of each figure, which means that the reader must guess what it refers to. The appropriate marking should be clearly specified in the figure or in the figure description

4. Line 24. There is Figure 6 should be Figure 7

5. Table 5 Table 5 should be redone, it is illegible, it would indicate that the most important are I0, i1, i2, i3, i4.

6. Figure 8. The symbols in the figure that are not legible are obscured by a blue and red line, please redo

7. Figure 9. Figures a and b are identical in my opinion, but they should be different

8. Line 303. There is Figure 8 (a) should be Figure 9 (a)

9. Line 310. There is Figure 8 (a) should be 9 (a)

10. Equaition 14 and 19. Once "cI" is used and once "Ic" is used please harmonize

11. Line 419. There is Figure 12 and it should be Figure 13

Author Response

Responses to Comments and Suggestions

Dear reviewer,

Thank you for your comments concerning our manuscript. The comments are all valuable and very helpful for revising and improving our paper, as well as the important guiding significance to our research. We have studied comments carefully and have made correction which we hope to meet with your approval.

We have made changes for the manuscript, as shown in the updated version. In the updated version, the changes are marked up using the “Track Changes” function (according to the editor's request). The following are reply to the reviewer:

Main comments:

Comment 1: Line 50. It's kw should be kW

Response 1: Thanks for the expert’s advice. We are sorry for making the mistakes. The kw has been modified to kW

Comment 2: Line from 149 to 154 there is a description of scr, and below there are no descriptions of other solutions, ie: src, rsc, rcs crs csr or a comment regarding the lack of such descriptions.

Response 2: Thanks for the expert’s advice. We modified the description of the different layouts in the manuscript. The six different layouts xyz(scr, src, rsc, rcs, crs, csr) can be described as: the output shaft of the engine is connected to x component; the pump-motor hydraulic mechanism of the HMCVT is connected to y component, and the output shaft of HMCVT (the input shaft of the rear axle) is connected to z component, respectively. Therefore, the design of src, rsc, rcs crs csr schemes is similar to scr. For example, src is defined as that the output shaft of the engine is connected to the sun gear; the pump-motor hydraulic mechanism of the HMCVT is connected to the ring gear, and the output shaft of HMCVT (the input shaft of the rear axle) is connected to the planet carrier, respectively. rsc is defined as that the output shaft of the engine is connected to the ring gear; the pump-motor hydraulic mechanism of the HMCVT is connected to the sun gear, and the output shaft of HMCVT (the input shaft of the rear axle) is connected to the planet carrier, respectively.

Comment 3: Figure 1. There is no description of the vertical bar on the right side of each figure, which means that the reader must guess what it refers to. The appropriate marking should be clearly specified in the figure or in the figure description.

Response 3: Thanks for the expert’s advice. The bar on the right in Figure 1 represents iout. Figure 1 has been modified to make the figure description clearer.(see word)

Figure 1. Result distribution graph of iout.

Comment 4: Line 240. There is Figure 6 should be Figure 7

Response 4: Thanks for the expert's advice. We are sorry for making the mistakes. It has been revised in the manuscript.

Comment 5: Table 5 should be redone, it is illegible, it would indicate that the most important are i0, i1, i2, i3, i4.

Response 5: Thanks for the expert's advice. Table 5 has been redone according to the comments, as follows:

Table 5. Gearbox gear pair transmission ratio and k value.

Parameter

Value

Parameter

Value

i0

1.01

i7

1.70

i1

1.32

i8

0.54

i2

0.58

i9

0.48

i3

1.89

i10

1.20

i4

2.12

i11

1.33

i5

1.97

k1

2.96

i6

1.91

k2

3.96

Comment 6: Figure 8. The symbols in the figure that are not legible are obscured by a blue and red line, please redo.

Response 6: Thanks for the expert's advice. To clearly represent the symbols in Figure 8, we move the symbols to appropriate positions beyond the blue and red lines. And the shades of the blue and red lines are lowered to make the picture clear,as follows:(see word)

Figure 8. Power flow in pure hydraulic section.

Comment 7: Figure 9. Figures a and b are identical in my opinion, but they should be different.

Response 7: Thanks for the expert's advice. We have checked the Figure 9. We are sorry for making the mistakes. The Figure 9 has been modified according to the engaging power method of the equivalent gear.(see word)

Figure 9. Equivalent transmission principle diagram: (a)HM1 (HM3); (b)HM2 (HM4).

Comment 8: Line 303. There is Figure 8 (a) should be Figure 9 (a)

Response 8: Thanks for the expert's advice. We are sorry for making the mistakes. Figure 8 (a) has been modified to Figure 9 (a).

Comment 9: Line 310. There is Figure 8 (a) should be 9 (a)

Response 8: Thanks for the expert's advice. Figure 8 (a) has been modified to Figure 9 (a).

Comment 10: Equaition 14 and 19. Once "cI" is used and once "Ic" is used please harmonize.

Response 10: Thanks for the expert's advice. The relationship between "cI" and "Ic" is iIc=1/ iCi, The symbol is uniformly modified to cI in equations 14 and 19.

Comment 11: Line 419. There is Figure 12 and it should be Figure 13.

Response 11: Thanks for the expert's advice. Figure 12 has been modified to Figure 13

The above are our team's answers to the experts' questions. Once again, we express our respect for the expert. We very much hope that these explanations will enable the expert and readers to further understand our research work.

    We tried our best to improve the manuscript and made some changes in the manuscript.

    Once again, we appreciate for reviewers’ warm work earnestly, and hope that the revised version will meet with approval.

Thank you and best regards.

Reviewer 3 Report

Some minor revisions highlighted in the manuscript.

Author Response

Responses to Comments and Suggestions

Dear reviewer,

Thank you for your comments concerning our manuscript. The comments are all valuable and very helpful for revising and improving our paper, as well as the important guiding significance to our research. We have studied comments carefully and have made correction which we hope to meet with your approval.

We have made changes for the manuscript, as shown in the updated version. In the updated version, the changes are marked up using the “Track Changes” function (according to the editor's request). The following are reply to the reviewer:

Main comments:

Thank reviewer for your comments and approval. The syntax errors in the manuscript have been corrected. Figure 3 and Figure 10 have been replaced with clearer figures.

Once again, we express our respect for the expert. We very much hope that these explanations will enable the expert and readers to further understand our research work.

    We tried our best to improve the manuscript and made some changes in the manuscript.

    Once again, we appreciate for reviewers’ warm work earnestly, and hope that the revised version will meet with approval.

Thank you and best regards.

Reviewer 4 Report

Concern 1#

The literature survey section must be expanded and the author is requested to cite the following latest paper  Gupta, B., Madan, G., & Md, A. Q. (2022). A smart agriculture framework for IoT based plant decay detection using smart croft algorithm. Materials Today: Proceedings.

Concern 2#

The author must include the problem statement and must clearly say what problem the paper is trying to solve

Concern 3#

The author must clearly explain the flow of work through the proposed architectural diagram. The diagram must have contents related to the detailed process

Concern 4#

 How have the outcomes been ensured in light of the major uncertainties?

Concern 5#

The conclusion has to be revised to incorporate the following advice: - Highlight your analysis and just present the most important takeaways from the full paper.

- Mention the advantages.

- In the final sentence of this section, mention the inference.

Make sure the topic of the Conclusion differs from what is discussed in the abstract.

- include the future work with more than one direction

Author Response

Responses to Comments and Suggestions

Dear reviewer,

Thank you for your comments concerning our manuscript. The comments are all valuable and very helpful for revising and improving our paper, as well as the important guiding significance to our research. We have studied comments carefully and have made correction which we hope to meet with your approval.

We have made changes for the manuscript, as shown in the updated version. In the updated version, the changes are marked up using the “Track Changes” function (according to the editor's request). The following are reply to the reviewer:

Main comments:

Comment 1: The literature survey section must be expanded and the author is requested to cite the following latest paper Gupta, B., Madan, G., & Md, A. Q. (2022).  A smart agriculture framework for IoT based plant decay detection using smart croft algorithm. Materials Today: Proceedings.

Response 1: Thanks for the expert’s advice. We checked the references and added the quotation of the article.(see word)

Comment 2: The author must include the problem statement and must clearly say what problem the paper is trying to solve.

Response 2: Thanks for the expert’s advice. This manuscript mainly provides a solution to the problem of low transmission efficiency of HMCVT for high-power tractors. According to the efficiency characteristics and speed ratio characteristics of single row HMCVT, the manuscript designed a double row HMCVT structure with optimal characteristics. The engine-HMCVT-load integrated modeling method was proposed, and the HMCVT hardware in the loop test platform was built, providing theoretical method support for efficiency improvement and analysis of high-power tractors.

Comment 3: The author must clearly explain the flow of work through the proposed architectural diagram.  The diagram must have contents related to the detailed process

Response 3: Thanks for the expert’s advice. The workflow of the manuscript is shown in Figure 1. First, six different single planet HMCVT structures are designed according to the transmission characteristics of planetary gears. Then the efficiency characteristics and speed ratio characteristics of different structures are analyzed to obtain the optimal transmission characteristic structure. According to the above results, the double row HMCVT with excellent transmission characteristics was designed and the hardware in the loop test platform of HMCVT was built. Finally, the engine HMCVT load integration scheme is proposed to analyze the full load efficiency characteristic field and variable load efficiency characteristic field.

Figure 1. Research roadmap(see word)

Comment 4: How have the outcomes been ensured in light of the major uncertainties?

Response 4: Thanks for the expert's advice. The biggest uncertainty in this study is the accuracy of the test. To solve this problem, we first calibrated the pump motor hydraulic test bench under different working conditions, as shown in Figure 2. Wherein, (a) is the comparative verification of simulation and test of motor output speed under constant displacement working condition, and (b) is the comparative verification of simulation and test of motor output speed under variable displacement working condition. Then the electric proportional valve of the clutch test bench was calibrated as shown in Figure 3. Finally, the two test benches were tested for many times to ensure the accuracy of the hardware in the loop test results.

Figure. 2 Comparison between test and simulation results of motor output speed (see word)

Figure. 3 Calibration test of clutch electro-hydraulic proportional valve(see word)

Comment 5: The conclusion has to be revised to incorporate the following advice:

 -Highlight your analysis and just present the most important take aways from the full paper.

- Mention the advantages.

- In the final sentence of this section, mention the inference.

- Make sure the topic of the Conclusion differs from what is discussed in the abstract.

- include the future work with more than one direction

Response 5: Thanks for the expert's advice. According to the comments, the conclusion is modified as follows:

1) The study proposes a mathematical method to define the layouts of the single row planetary HMCVT. This method uses the planetary gear characteristic parameter k as the independent variable to combine the three parts of the planetary gear with the mechanical system which can obtain different layouts. According to the transmission ratio characteristics and efficiency characteristics of different layouts, the suitable layouts are selected to de-sign a double planetary HMCVT, which is designed with optimal transmission efficiency and gear ratio range.

2) Based on the characteristics of hydraulic power split ratio, mathematical model, pump motor hydraulic test bench and clutch system test bench, the hardware-in-the-loop system of HMCVT is designed, and the engine-HMCVT-load is built as an efficient whole.

3) The efficiency characteristic field in maximum load and variable load (5-11km/h) are designed and analysed respectively. The results show that the efficiency characteristics of HM2 and HM3 are better in full load, so the speed range should be controlled at this stage as much as possible during operation .The transmission efficiency of HM2 with power split is lower than that of HM1 at the shift point, so the shift is generally at the low gear.

4) This study completed the design part and the characteristic analysis part. However, the hardware in the loop test of HMCVT is the first step. In the future, a control algorithm needs to be added to control the TCU of HMCVT to test the reliability of HMCVT shifting. Then further real vehicle test is required to verify the designed HMCVT.

The above are our team's answers to the experts' questions. Once again, we express our respect for the expert. We very much hope that these explanations will enable the expert and readers to further understand our research work.

    We tried our best to improve the manuscript and made some changes in the manuscript.

    Once again, we appreciate for reviewers’ warm work earnestly, and hope that the revised version will meet with approval.

Thank you and best regards.

Round 2

Reviewer 1 Report

I wish to thank the authors for addressing all my questions. I just recommend a final check for possible misprints.

Author Response

Responses to Comments and Suggestions

Dear reviewer,

Thank you again for your comments concerning our manuscript. The comments are all valuable and very helpful for revising and improving our paper, as well as the important guiding significance to our research. We have studied comments carefully and have made correction which we hope to meet with your approval.

We have made changes for the manuscript, as shown in the updated version. In the updated version, the changes are marked up using the “Track Changes” function (according to the editor's request). The following are reply to the reviewer:

Main comments:

Comment: I wish to thank the authors for addressing all my questions. I just recommend a final check for possible misprints.

Response 1: Thanks for the expert’s advice. The manuscript was carefully checked and revised, which were marked up using the “Track Changes” function. In particular, the formatting of figures and words has been carefully adjusted to ensure compliance with journal's requirements.

Once again, we express our respect for the expert and we tried our best to improve the manuscript and made some changes in the manuscript.

   We appreciate for reviewers’ warm work earnestly, and hope that the revised version will meet with approval.

Thank you and best regards.
